# Machine Learning Application Using Cost-Effective Components for Predictive Maintenance in Industry: A Tube Filling Machine Case Study

David Natanael and Hadi Sutanto *

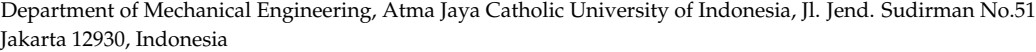

Department of Mechanical Engineering, Atma Jaya Catholic University of Indonesia, Jl. Jend. Sudirman No.51, Jakarta 12930, Indonesia
* Correspondence: hadi.sutanto@atmajaya.ac.id

**Abstract:** Maintenance is an activity that cannot be separated from the context of product manufacturing. It is carried out to maintain the components' or machines' function so that no failure can reduce the machine's productivity. One type of maintenance that can mitigate total machine failure is predictive maintenance. Predictive maintenance, along with the times, no longer relies on visuals or other senses but can be combined into automated observations using machine learning methods. It can be applied to a toothpaste factory with a tube filling machine by combining the results of sensor observations with machine learning methods. This research aims to increase the Overall equipment effectiveness (OEE) to 10% by predicting the components that will be damaged. The machine learning methods tested in this study are random forest regression and linear regression. This study indicates that the prediction accuracy of machine learning with the random forest regression method for PHM predictive is 88% of the actual data, and linear regression has an accuracy of 59% of the actual data. After implementing the system on the machine for three months, the OEE value increased by 13.10%, and unplanned machine failure decreased by 62.38% in the observed part. Implementation of the system can significantly reduce the failure factor of unplanned machines.

**Keywords:** manufacturing; predictive maintenance; machine learning; OEE

## 1. Introduction

Maintenance is a combination of activities to restore a component or a machine to a state where the component/machine can work according to its initial function [1]. Maintenance management is essential for process efficiency, maximizing profits with minimized hidden costs [2]. One crucial maintenance element in the smart manufacturing application is prognostic and health management (PHM) [3] for identifying failure in critical parts of the machine so that preventive maintenance can take place [4]. PHM is an application of predictive maintenance (PdM) [5]. PdM is a PHM form based on continuous observation of parts/machine conditions [6]. Predictive maintenance has already evolved from visual observation by humans to automated observation, signal processing, pattern recognition [7], machine learning [8], neural network [9], fuzzy logic [10], and many other methods. The visual observation method cannot give a total prediction result [11]. The automated observation method using sensors becomes the solution to provide and collect raw data/signals from parts/machines for real-time monitoring [12]. Compared with PdM, a common maintenance management strategy is scheduling maintenance actions to avoid failures [13]. This method can be combined with collected data from condition monitoring (CDM) and condition analysis algorithm to make autonomous predictive maintenance [14]. Predictive maintenance activity can be classified into [15]: (1) database approach, (2) model approach, and (3) hybrid approach. For applied PdM in the toothpaste industry, the main machine can be classified as [16]: (1) mixing machine and (2) filling machine. It will be combined with historical data from the observed component to make a predictive analysis system [17].

When analyzing the data collected through various observation sensors, machine learning in dataset processing has a high accuracy [18] and can predict various machine conditions [19]. PdM in the toothpaste factory will be applied in the tube filling machine, which consists of many crucial sections [20]: (1) tube infeed—for packaging materials input using a set of pneumatics and electrical motors, (2) tube orientation—for tube positioning using a servo motor and eye mark sensor to detect the eye mark in tubes, (3) tube filling—to input the toothpaste from hopper using piston mechanism and filling pump mechanism, (4) hot air station—heating the tube using blow of hot air that has been heated using spiral wire heater cartridge, (5) tube sealing—a set of sealing jaws and knife, to seal the tube and trim the excess material from the tube, and (6) pick and place unit—sealed tube output. The PdM will be applied using a combination of machine learning (hybrid) methods. The tube filling machine figure with each of the functions is shown in Figure 1.

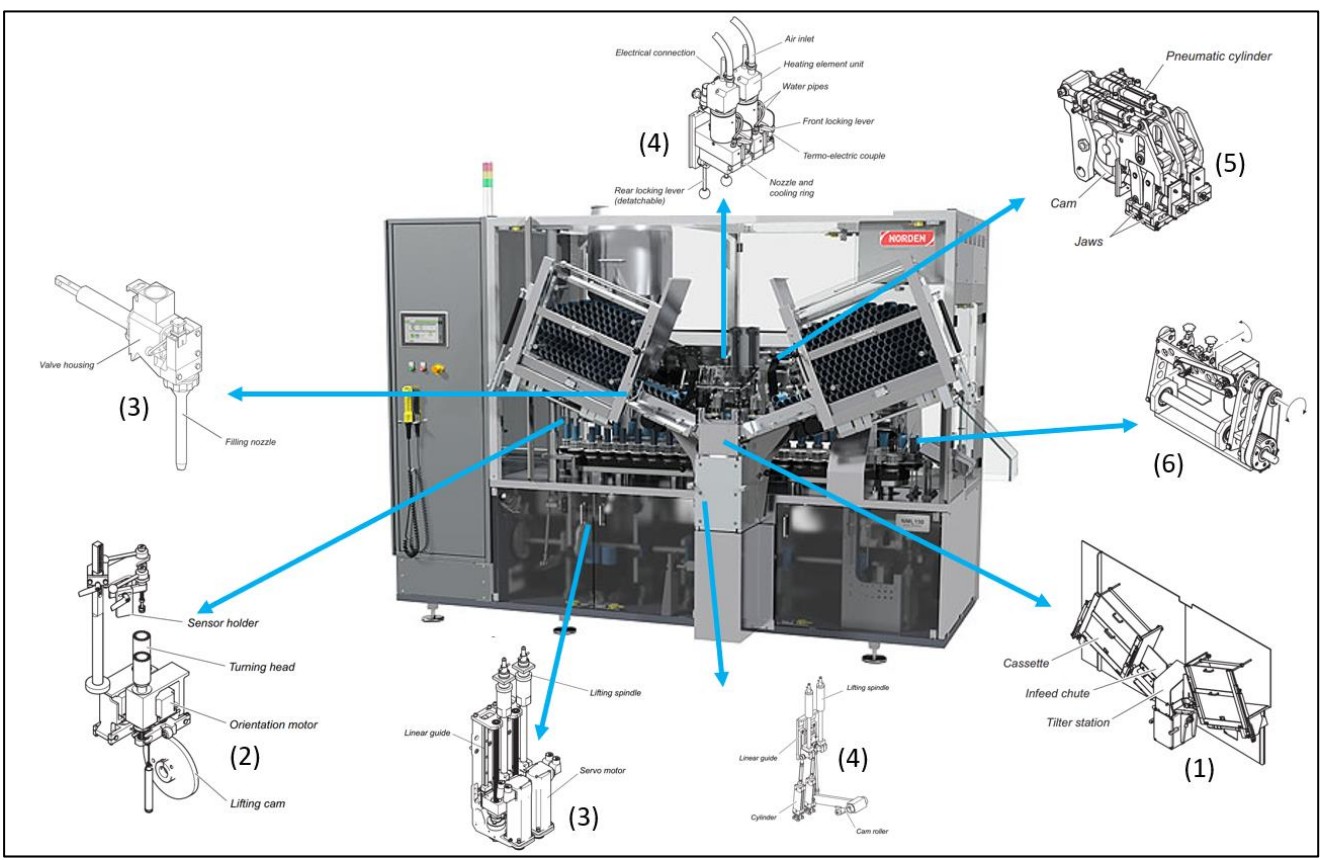

**Figure 1.** Tube filling machine with component list: (1) Tube infeed, (2) Tube orientation, (3) Tube filling, (4) Hot air station, (5) Tube sealing, (6) Pick and place unit [20].

From those parts, if one part is unable to function due to failure, the machine will not be able to operate, or it will operate with a lot of rejected products [20]. The machine can have several critical parts that will affect production output and machine breakdown if those vital parts do not work correctly [21]. When the machine cannot produce good output or has a breakdown, it will affect plant and machine efficiency [22]. A method called overall equipment effectiveness (OEE) can be used to measure the effectivity of a machine with three scales [23]: (1) Availability rate (AR), (2) Performance rate (PR), and (3) Quality rate (QR). To prevent these events, a new PdM method combined with a machine-learning algorithm predicts future data using a supervised method from past datasets or initial dataset movements [24]. It compares the database with the operation data set. Research to improve OEE in machinery was also done by Brunelli et al. in the automatic filling machine, using deep learning method with temporal convolutional network (TCN) and comparing it with the long short-term memory (LSTM) method [25]. Additionally, another research by

Paolanti et al. was done for predictive maintenance in woodworking machinery using a random forest approach [19]. Borgi et al. also researched predictive maintenance using the multiple linear regression approach for industrial robots [26]. The Ishikawa diagrams used to identify the research can be seen in Figure 2.

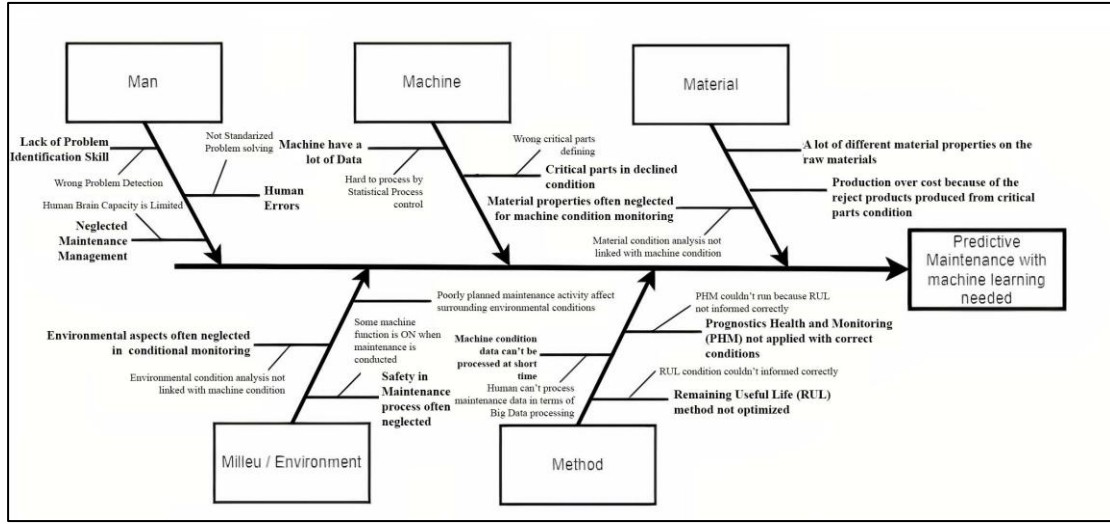

**Figure 2.** Fishbone diagrams for the research background [27–30].

Some reasons for the research can be seen in Figure 2, the need for predictive maintenance using machine learning. Dalzochio et al. propose the problem from a machine perspective in the diagram [27]. From the man's perspective, the need for predictive maintenance using machine learning is proposed by Binding et al. [28]. Environmental root causes are researched by Nacchia et al. [29], and materials root causes are stated in the research of Bampoula et al. [30]. Due to some limitations (e.g., company rules and budget), there is some constraint applied in this research:

i.   The system made was a prototype that was chosen using cost-efficient materials;
ii.  The system will use two supervised machine learning methods to compare the effectivity of random forest regression and linear regression;
iii. The system will trace failure based on a failure that made the product rejected or downtime;
iv.  The system will ignore a failure that consists of human error;
v.   The machine is operating for a maximum of 16 h a day.

The consideration is to compare two supervised machine learning methods, linear regression and random forest regression [31]. Linear regression approaches are chosen because the mechanism of the analyzed parts only has 2 degrees of freedom (DoF) at maximum from the observed parts (either X, Y, or Z axis)—this means that the extracted electrical value only has one independent predictor. Borgi et al. also proposed to analyze robot movements using the multiple linear regression method because the extracted data consists of more than one independent/predictor available [26]. The random forest approach is chosen because it is a series of tree predictors in which each tree is based on the values of a randomly sampled vector with the same distribution across all trees in the forest [32]. The application in temperature-related research was also made by Prihatno et al. in predicting humidity in industrial factory applications [32]. Other predictive maintenance research was done by many researchers, such as Paolanti et al., that did predictive maintenance in woodworking machinery using a random forest approach, but with a flight recorder, an industrial class PC, an IT accelerometer, and an industrial class accelerometer [19]. Similar research was also done by Borgi et al., using an LTD800 Leica Laser Tracker as a sensor for tracking robot movements in 6 DoF [26]. The main goal of this research is to predict the time to failure of each observed component by feature extraction and machine learning. It also

measures each system's effectiveness and weakness to seek the most applicable method for vibration and temperature measurement [33] in tube filling machine applications because every machine part has its differences and characteristics, even for the exact variable measurement [34]. Another contribution of this research is to apply predictive maintenance using machine learning with cost-effective components as the primary condition.

## 2. Materials and Methods

### 2.1. Experimental Setup

The PdM system will implement in NORDEN NML—150 tube filling machine, which has a specification in Table 1.

**Table 1.** Norden NML-150 tube filling machine specification [20].

| Parameter | Value | Unit |
|---|---|---|
| Machine Speed | 150 | rpm |
| Nozzle Head Count | 2 | pcs |
| Machine Power | 8.20 | kWh |

With the machine specification as in Table 1, the historical data of unplanned downtime (UPDT) from the last six months in 2021 can be seen in Figure 3.

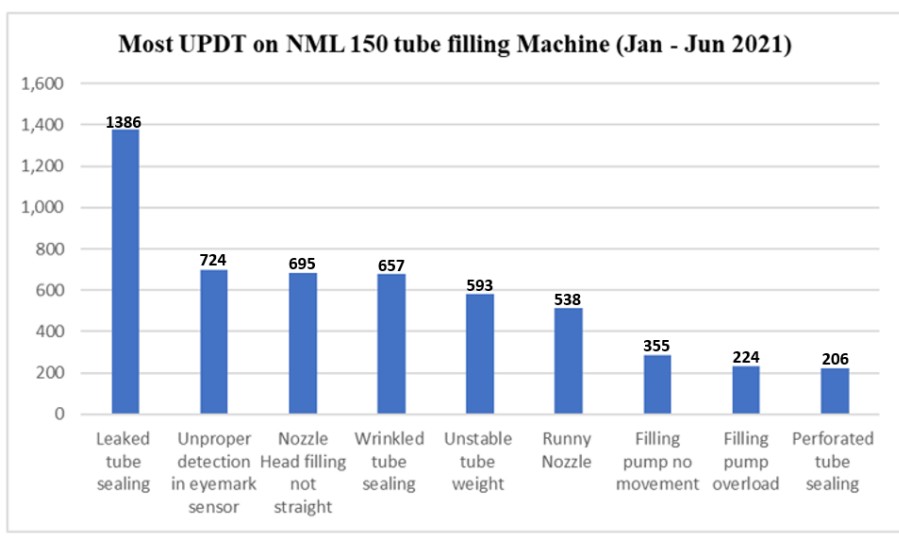

**Figure 3.** Machine historical data of unplanned downtime for six months (January 2021 – June 2021).

Figure 3 shows the 10 most common breakdowns of the machine, where 6 out of 10 UPDT lower the OEE percentage in terms of availability rate (AR) and 4 other downtimes lower the quality rate (QR). The downtime occurred in the machine with a low frequency (less than 3 times/category)—and high repair time. The explanation for the downtime with a low frequency is because the downtime affected critical parts, such as bearings and bushings, which significantly affected the mechanical movement of the machine and affected production results (reject products mean a deficit for the company). The total breakdown frequency from Figure 3 is 16 times, which means each breakdown has an average of 336.125 min UPDT. Each breakdown can be sorted and divided into several specific machine parts that can be observed, where the classification is shown in Table 2.

After classifying UPDT and analyzing the affected parts of the machine, it is noticeable that there are two or more UPDT cases in one part that needs to be further organized to decide which parts need to be further analyzed and to solve the problems when the microcontroller ports to analyze these components are limited. Depending on the affected parts, the amount of the sensor will be analyzed based on how many downtimes were involved [20]; the component classification can be seen in Table 3.

**Table 2.** Downtime analysis and affected parts classification [20].

| Downtime Assessment | Main Problem | Affected Parts Analysis |
|---|---|---|
| Leaked Tube Sealing | Un-perfect sealing position, toothpaste leaks when pressed. | Hot Air unit—Thermocouple Lifting unit—Servo Motor Coding Cam—Vibration sensor |
| Improper detection in eye mark sensor | Unsymmetrical sealing position, tube positioning in eye mark sensor. | Tube orientation—photocell sensor Tube orientation—servo motor |
| Improper filling position | The lifting position on tube filling is not in a straight line. | Lifting unit—servo motor Filling pump unit—vibration |
| Wrinkled tube sealing | Imperfect sealing position, thin line in the seal, toothpaste leakage from the seal | Hot Air unit—Thermocouple Coding Cam—Vibration sensor |
| Runny Nozzle | The filling nozzle cut-off is not perfect and affects the sealing position in hot air and other section | Filling pump unit—vibration Filling pump drive—vibration |
| Filling pump no movement | Mechanical movement from the filling pump stopped affects machine operation. | Filling pump unit—vibration Filling pump drive—vibration |
| Filling pump overload | Over-stroke in mechanical movement affects machine operation. | Filling pump drive—vibration |
| Perforated tube sealing | Imperfect tube sealing, huge cracks in sealing position, toothpaste leakage from the seal | Hot Air unit—Thermocouple Coding Cam—Vibration sensor |

**Table 3.** Component classification table from filling machine NML 150 [20].

| Component—Measuring Units | Amount | Downtime Assessment |
|---|---|---|
| Temperature Sensor—Thermocouple | 2 | Leaked tube sealing Perforated tube sealing |
| Lifting unit—servo motor | 2 | Leaked tube sealing Improper filling position |
| Coding cam—vibration sensor | 2 | Leaked tube sealing Wrinkled tube sealing Perforated tube sealing |
| Tube orientation—photocell sensor | 2 | Improper detection in eye mark sensor |
| Tube orientation—photocell sensor | 2 | Improper detection in eye mark sensor |
| Filling pump unit—vibration sensor | 4 | Improper filling position Filling pump no movement Filling pump overload Unstable tube weight |
| Filling pump drive—vibration sensor | 2 | Unstable tube weight Filling pump no movement Filling pump overload |
| Filling pump unit—filling pump overload proximity | 2 | Filling pump overload |

From Table 3, the **component—measuring units** is the component names and its sensor to enable data acquisition; the **amount** is the amount of the components in one machine, and the **downtime assessment** is the classified downtime from Table 2, classified by the components.

### 2.2. Components and Sensor Placement

From the component classification, the observed area will be the filling pump unit using 4 vibration sensors because 4 downtimes occur. The coding cam—uses 2 vibration sensors because 3 downtimes are occurring in the area. The filling pump drive—uses 2 vibration sensors. After all, 3 downtimes occur in the area and the temperature sensor or

thermocouple. The critical parts have been decided, then the system used will be chosen using cost-efficient materials. The microcontroller used is Arduino AT Mega 2560 with 16 Analog ports. The thermocouple sensor is a type K thermocouple with a maximum temperature range from 0 to 700 °C. In contrast, the vibration sensor will use an accelerometer ADXL–335 with nine axes reading and sensitivity rate between 270–320 mV/g and 1600 Hz Bandwidth. Figure 4a–d shows the position of all sensors.

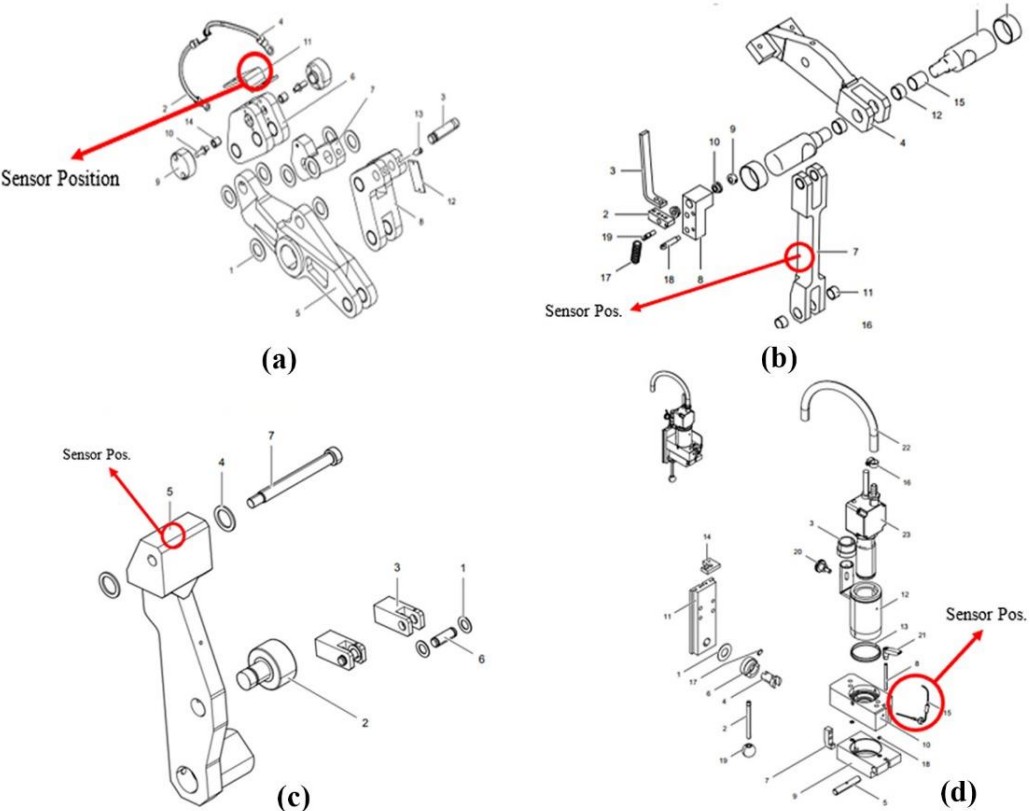

**Figure 4.** Sensor placement on tube filling— (**a**)filling pump drive, (**b**) filling pump lever, (**c**) coding cam lever, (**d**) hot air station, thermocouple [35].

In Figure 4a, the sensor applied is a vibration/accelerometer sensor for detecting filling pump movement; this part's function is to transfer toothpaste from hopper to nozzle [35]. In Figure 4b, the vibration/accelerometer sensor is applied to detect filling pump movement from the bottom of the machine. This part's function regulates toothpaste weight using piston movement linked to shaft and bushings [35]. Figure 4c shows that the sensor is a vibration or accelerometer to detect coding cam movement. This part's function is to seal a filled tube that has been heated in a hot-air station and give its expiration date [35]. In Figure 4d, the sensor applied is a thermocouple sensor to detect hot air temperature and regulate the temperature in hot air stations [35]. The location of the actual sensor's position can be seen in Figure 4.

Electrical connections—from the microcontroller, control system, and sensors can be seen in Figure 5.

From the electrical diagrams in Figure 6, the ports used for data acquisition are analog ports for analog-type transducers (thermocouples and accelerometers). In addition, the transducer's power source comes from a 5 VDC power supply to ensure the sensors have a suitable electrical supply with the sensor's specification. The number of ports used in Arduino for data acquisition is 14 analog ports. The breadboard is used for interconnections between the electrical supplies and grounds.

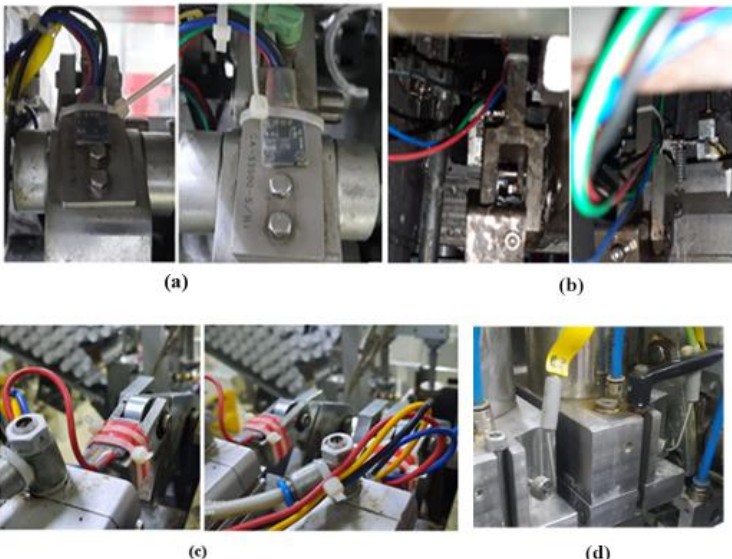

**Figure 5.** Actual sensor's position in the machine (**a**) filling pump drive, (**b**) filling pump lever, (**c**) coding cam lever, and (**d**) hot air station, thermocouple.

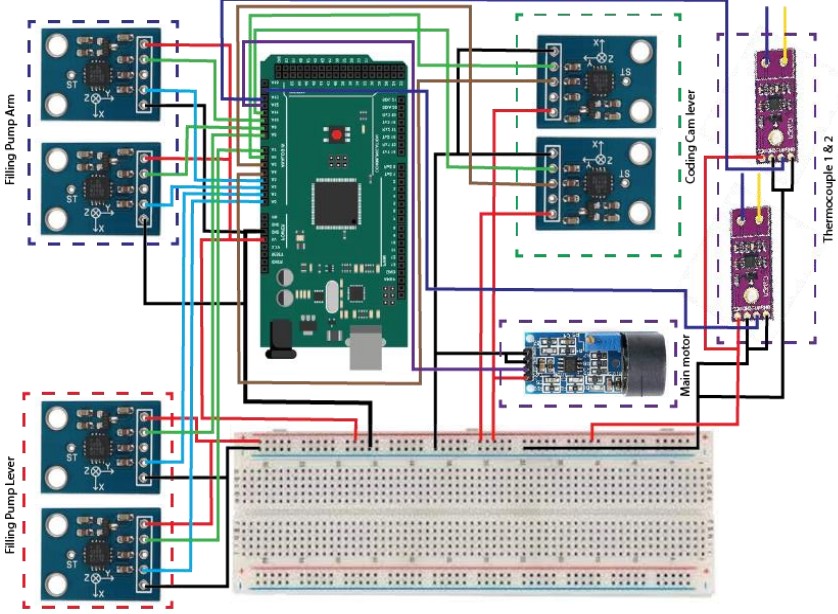

**Figure 6.** Electrical connections on the sensors and microcontroller.

### 2.3. Method and Algorithm

The flow diagram of data logging and the machine learning method is shown in Figure 7.

The flow diagram shows that the system is started from a data logging mechanism. The current sensor value is combined with historical data logging data. Its output value will be in the serial monitor from the Arduino data log. The serial monitor from data logging has a 9600 baud rate connection and sampling time of 1 s for each data. A separate server is used to process acquired data using a Python programming language for data logging progress. The data logging algorithm will explain the data as seen in Algorithm 1, where $n$ is the data counts, $X^{ax}$ is the X-axis sensor ports, $Y^{ax}$ is the Y-axis sensor ports, and $Z^{ax}$ is the Z-axis sensor ports. The $C^x$, $C^y$, and $C^z$ value is a calibrated value of $X$, $Y$, or $Z$-axis ports. $N$ is the number of ports used. The data acquired will be in a standard format of comma-separated value files and will have an array for further processing.

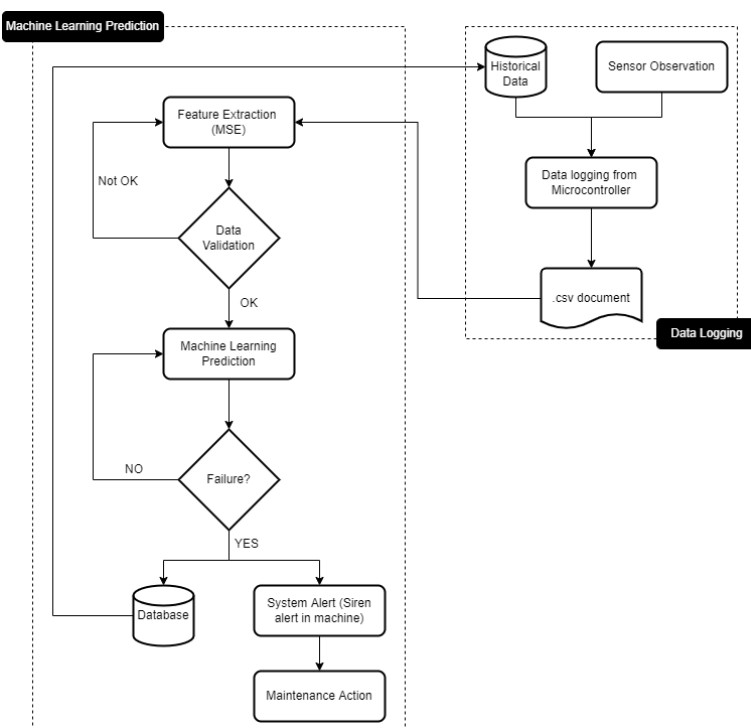

**Figure 7.** Flow diagram of data logging and predictive maintenance using machine learning.

---

**Algorithm 1.** Data Logging Algorithm

---

| **Require** | : | $n = 0$ | ▶ Starting data count from 0 |
|---|---|---|---|
| **Ensure** | : | $X^{ax} = C^x_{val}$ , $Y^{ax} = C^y_{val}$ , $Z^{ax} = C^z_{val}$ | |
| | | $Min\ C^x_{val}\ \leq\ X^{ax}\ \leq\ Max\ C^x_{val}$ | |
| | | $Min\ C^y_{val}\ \leq\ Y^{ax}\ \leq\ Max\ C^y_{val}$ | ▶ Ensure the value between Max and Min |
| | | $Min\ C^z_{val}\ \leq\ Z^{ax}\ \leq\ Max\ C^z_{val}$ | |
| | | **While** $n \geq 0$ **do** | |
| | |   **If** $n$ is running **then** | |
| | |     $.csv \leftarrow XN\ x\ YN\ x\ ZN$ | ▶ N = Number of Sensors Port |
| | |   **Else if** $n$ is stop **then** | |
| | |     $.csv = 0$ | ▶ .csv stop working |
| | |   **End if** | |
| | | **End While** | |

---

From Algorithm 1, .csv data will be acquired using a data logging program in Python. Further note, the .csv data must have fulfilled some of the conditions below:

i. Spacing for each column will be separated with a comma (default);
ii. The decimal value will use a dot (.) to specify the value;
iii. The timestamp algorithm will use a dash (-) separator.

To process data in machine learning, before the machine learning algorithm is implemented, we need a training dataset to train our data model, both in linear regression or in the random forest regression method. The sampling amount of data used to train the value can be seen in Table 4.

Note for Table 4, **FPL** is a filling pump lever, **FPD** is a filling pump drive, **CCL** is a coding cam lever, and **Th. Coup** is a thermocouple sensor. The decision to use small training data came with inspiration from Paolanti et al. Their research stated that the ML algorithm used a random forest and decision tree, with a total of 530,731 data from 15 machines [19], which means 1 machine contributed 35,382 data on average. Borgi et al. also used small sampling data for predictive maintenance in industrial robots, which has a similar task

regarding accelerometer sensor data acquisition. This results in excellent accuracy in mean square error (MSE) and root mean square error (RMSE) using the linear regression learning method [26]. The sampling amount was also decided because of restriction number 5—in the Introduction section—which tells the short period of maximum running time (16 h a day) and the faster system implementation on the machine. After deciding on the sampling amount from the data logging, then the algorithm of machine learning prediction can be seen in the machine learning prediction algorithm, which can be seen in Algorithm 2.

**Table 4.** Sampling data amount on each sensor.

| Sensor Name | Sample Amount (Data) | Sensing Time (min) |
|---|---|---|
| Acc. FPL 1 | 20.000 | 57.143 |
| Acc. FPL 2 | 20.000 | 57.143 |
| Acc. FPD 1 | 20.000 | 57.143 |
| Acc. FPD 2 | 20.000 | 57.143 |
| Acc. CCL 1 | 20.000 | 57.143 |
| Acc. CCL 2 | 20.000 | 57.143 |
| Th. Coup 1 | 20.000 | 57.143 |
| Th. Coup 2 | 20.000 | 57.143 |

---

**Algorithm 2.** Machine Learning Prediction Algorithm

**Require** : $x \geq 0$　　　　　　　　　▶ There must be a data transfer process
**Ensure** : $x \neq 0$
**Ensure** : $n \geq 0$　　　　　　　　　▶ Number of repetition must $\geq 0$
　　　　**While** $n \geq 0$ **do**
　　　　**If** $n$ is running **then**
　　　　$Nm =$
　　　　$\begin{pmatrix} x_n & x_{n-1} & x_{n-2} & x_{n-3} \\ x_{n-1} & x_{n-2} & \ldots & x_{n-m} \end{pmatrix}$
　　　　　**If** $NaN$ is available **then**
　　　　　　$NaN \leftarrow reshape = 0$
　　　　　　$NaN \leftarrow 0$
　　　　　　$Nm \leftarrow Array - 3$
　　　　　**End if**

　　　　　$\hat{f}_{rf}^K(Nm) = \frac{1}{K} \sum_{K=1}^{K} T(Nm)$　　　▶ Random forest reg. k = Rand State
　　　　**Else If** $n$ is stop **then**
　　　　　$Nm = 0$　　　　　　　　　▶ .csv stop working
　　　　**End if**
　　　　**End While**

---

The measured values from the data logging process affected the whole system from this classification. The data is separated into eight different sections, and each section has its input and output of which the input can be defined as position change in the accelerometer sensors [19] and temperature change in the thermocouples [28]. The expected output from the experiment is a regression and regression prediction within two different models [28]. The parameters/value are interval data logging time and delay time from processing data. Beddows and Mallon stated that Arduino's delay time in the data logging interval could differ for each sensor [36]. There was also a delay time for data processing—from the data transfer time [37] and data processing time [38]. The parameters are shown in Table 5.

The parameter of delay transfer PC to Python and delay prediction to real-time affects the experimental results in which the data predicted is 0.23 s slower than the actual value. The response is created because of the transfer time between devices and algorithm time to calculate each acquired data from Python software. Machine learning data uses time-domain analysis because it is simplified, and the data accuracy can reach up to 93.8% [33].

**Table 5.** Time parameter for data sampling.

| Parameter | Value | Units |
|---|---|---|
| Data Logging Interval (Vibration) | 110 | ms |
| Data Logging Interval (Thermocouple) | 95 | ms |
| Delay Transfer PC To Python | 10.25 | ms |
| Delay Prediction to Real-time | 220 | ms |

The design of experiment (DOE) uses a random forest regression and linear regression method for movement or temperature prediction. For the random forest tuning—there are some tuning parameters based on the random forest equation [39]:

$$\hat{f}^K_{rf}(x) = \frac{1}{K} \sum_{K=1}^{K} T(x) \tag{1}$$

where *x* is the input vectors, made up of the value from different evidential feature analyses, and *K* is the number of regression trees in the equation and averages the results. The training data set is validated by fitting training data with the actual data and measuring errors from the training data. The several steps to doing the data validation process include:

1. Feature extraction from the sample data using C++ open-source programs [39] by eliminating the noise from the acquired data (sensor position change and temperature change) from the data acquisition program;
2. Data training using data from Table 4 and fitting data into the random forest and linear regression separately, with a total of 16 datasets, having been trained;
3. From the total of 16 datasets, the optimal *hyper-parameters* are found using cross-validation of the k-sections method [19]. The method will randomly subdivide the examples data into "k" sections, and for each value of parameters, the learning algorithm is executed for "k" times [19]. For the best results, hyperparameters were used in the experiment, such as *sample split* 10, *estimators* 5500, and *random state* 40 (for random forest prediction). The results also agree with the research of Prihatno et al. in terms of humidity predictions [32];
4. For better prediction results, means square error (MSE) and root mean square error (RMSE) are also calculated and fitted from 16 samples. The MSE and RMSE values are used to compare the training data accuracy with the real system data accuracy [40]. The results of MSE and RMSE from the training data can be seen in Table 6.

**Table 6.** MSE and RMSE result from training data.

| Training Data | MSE LR | RMSE LR | MSE RFR | RMSE RFR |
|---|---|---|---|---|
| Coding Cam Lever 1 | 0.060796765 | 0.24657 | 0.048271 | 0.219707 |
| Coding Cam Lever 2 | 0.05242268 | 0.22896 | 0.037664 | 0.194072 |
| Filling pump Lever 1 | 0.05547909 | 0.23554 | 0.022873 | 0.1511238 |
| Filling Pump Lever 2 | 0.04693722 | 0.21665 | 0.045772 | 0.213944 |
| Filling Pump Drive 1 | 0.08900079 | 0.29833 | 0.034782 | 0.186499 |
| Filling Pump Drive 2 | 0.07666807 | 0.27689 | 0.029887 | 0.172879 |
| Thermocouple 1 | 0.0728892 | 0.26988 | 0.038825 | 0.197041 |
| Thermocouple 2 | 0.06636291 | 0.25671 | 0.032419 | 0.180053 |

From the data acquisition and machine learning algorithm, data characteristic and prediction value of data is acquired and stated in Section 3.

## 3. Results and Discussion

### 3.1. Machine Condition Monitoring

From both machine learning and data logging algorithms, the acquired condition from analyzed components can be classified as:

i.   Normal acceleration/vibration condition;
ii.  Normal temperature condition;
iii. Run to Fail acceleration/vibration condition;
iv.  Run to fail temperature condition.

The analyzed conditions in the component were decided by reference to another research about remaining useful lifetime (RUL), where Lei et al. stated that all of the machinery components have a health indicator (HI) and all machinery components have at least two or more HI stages [41]. In the experiment, each component's HI has two stages—normal and run to fail (faulty).

The first result in Figure 8a shows the graph from a normal condition on implemented acceleration/vibration sensor in the machine component. The graphs in Figure 8 represent all conditions from acceleration/vibration cycles because the component movement characteristic is a cyclic movement [42]. Y-axis in the data explains sensor movement from zero position, and the movement on the sensor will have a repetition, from a positive to a negative position. Each cycle occurs in 2 s (from the graph, each upper peak vs. each lower peak, the change occurs every 1 s). The acceleration/vibration position range in one minute is 0.34 mm–1.05 mm/movement, and with the cycle model, the average action in a 1-min cycle time is 0.64 mm. If the direction of the component has a different pattern / outside this range, then the element is in a run-to-fail state, or there is function degradation in analyzed components. Figure 8b shows an acceleration/vibration data sample at the end of a component life cycle. There is a significant acceleration/vibration position increase from the component (in millimeter-scale—on Y-axis). Where the initial data should be in the form of a cycle, there is movement change into a rising movement in a cycle (red cycle) indicated by a change in data distribution model, from data range −0.5 mm to 0.5 mm to data range 1 to 2.5 mm. Machine stop is operated to run on predictive maintenance before experiencing a severe failure.

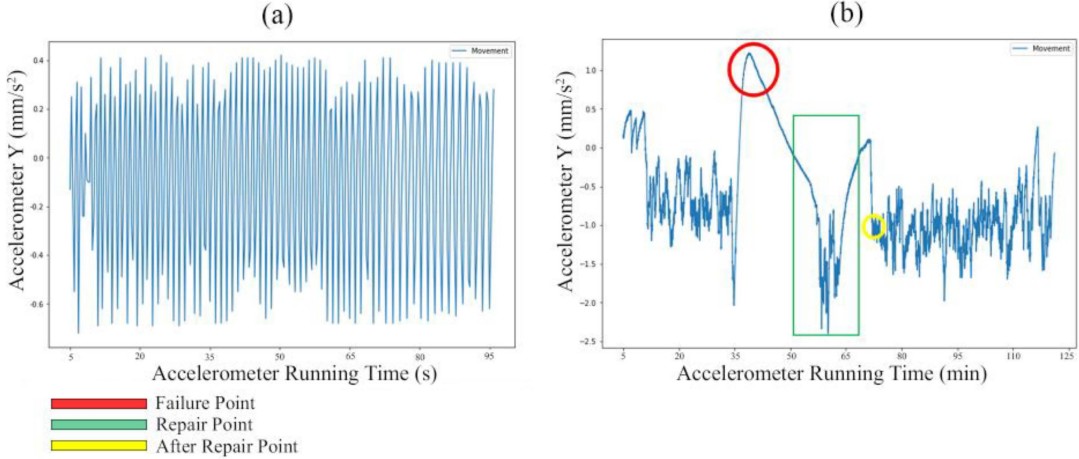

**Figure 8.** Graphs of sample cyclic vibration data in (**a**) normal condition, (**b**) failure condition.

From the thermocouple or temperature sensor, the data sample of normal function thermocouples is shown in Figure 9.

From Figure 9a, the normal temperature in the cycle ranged between 164–165 °C, where the initial setting for the hot air station is 165 °C. From the data sample, the characteristic of the hot-air station is more dominant to have a temperature decrease of +/−1 °C, where the temperature does not impact the tube sealing results. The graphic can be expanded into a table with a 1 s interval based on evaluating the range and tolerance in normal temperature data. From this table, the range of the hot air heater temperature in 1 min is 0.02–1.93 °C, with an average temperature of 0.69 °C. If there is movement above 1.93 °C, it can be said that the heater is in a run-to-fail mode or there is a function degradation. The mechanical component that runs to fail or at the end of its life can be seen in Figure 9b. Sample data

in the green rectangle shown in Figure 9b indicated the component change process until the component was rerun in a yellow cycle. From the graph, some range is higher than the average normal range at the end of the graph. The component configuration is not normal (human failure in the setting process).

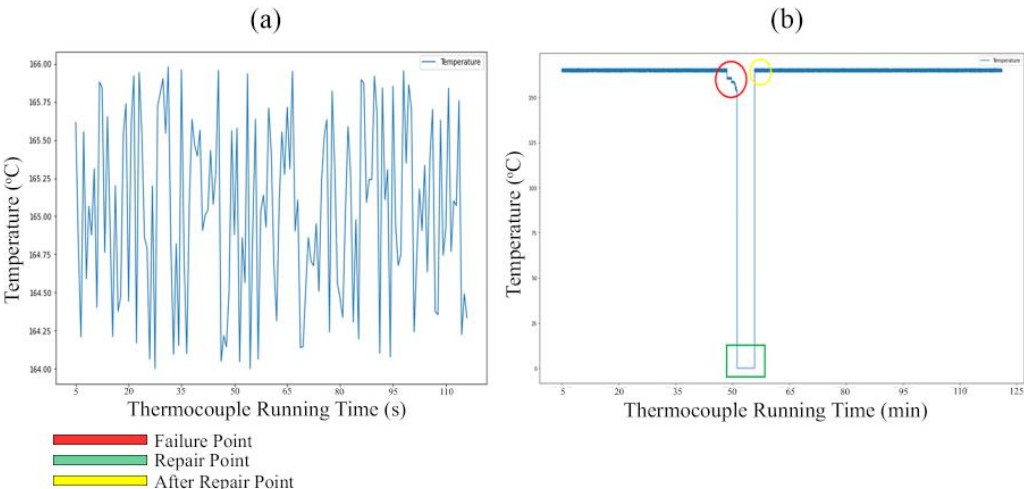

**Figure 9.** Graphs of sample temperature data in (**a**) normal condition, (**b**) failure condition.

Sample data of the run to fail heater in temperature value can be seen in Figure 8b. Heater failure starts from the value decrease in the heater that is marked by a red circle in the graphic. Thermocouple enhancer data samples showed that the heater has an exponential reduction from 163 °C in a few minutesAfter that, the temperature gradually decreased exponentially from 160 °C to 150 °C, which affected product quality in machine operation. The repair/replacement process can be seen in the green rectangle mark on the graphic, and the new heater operates at the yellow circle mark on the graphic. For encountering the outlier values from the thermocouple, the initial value was manually set when the system started, as well as the upper and lower boundary (Approx. ±2 °C). After acknowledging the characteristic of machine condition monitoring, machine learning prediction was implemented in each data, and it is discussed in Section 3.2.

### 3.2. Machine Learning Prediction Value

Data from Section 3.1. are treated as training datasets or initial datasets to make predictions using two different approaches in machine learning, random forest regression prediction (RF), and linear regression prediction (LR) [43]. The results of vibration and temperature data types are dynamic—based on the cited research [44]. The data prediction can be classified as follows:

1. RF Regression prediction using normal condition;
2. RF Regression prediction using failure condition;
3. LR prediction using normal condition;
4. LR prediction using failure condition.

The graphs are shown using vibration or accelerometer data because the model of temperature data is the same as the vibration data model with the same purpose because the system is evaluated using a knowledge-based (supervised machine learning method) [45]. However, the thermocouple cycle runs faster than accelerometer data. McLoone et al. state that measuring accelerometers placed in extreme temperatures using six thermocouples can send more frequent data than accelerometers [46]. Samples were collected from 26 cycles when it almost represented the data model in a 1 min cycle (refer to the normal condition data collection) [47]. The data from machine learning prediction (RF and LR) in the normal cycle condition of the monitored component is shown in Figure 10.

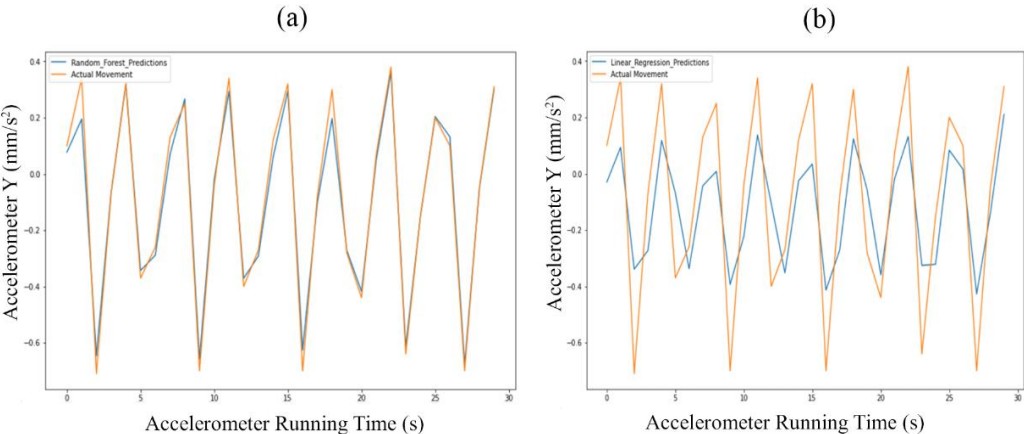

**Figure 10.** Data prediction in normal conditions using (**a**) RF regression method, (**b**) LR regression method.

From Figure 10a, the graphic shows data prediction using RF regression prediction. In a normal condition, data accuracy ranges between 40 and 99%, while the total prediction accuracy is approximately 82%. The data above are samples from 1 min data, so approximation inaccuracy may differ between actual and sample data. The prediction data from LR Regression prediction using the normal cycle condition of the monitored component can be seen in Figure 10b. Using LR prediction, the range of data accuracy differs between 3% and 80%, where the total data prediction accuracy is approximately 23% for 26 data samples. This method cannot predict cycle data because the data model is not linear, so the accuracy prediction is low. This method cannot be modified using data diversification, such as the RF regression prediction method. The LR data model does not have a decision tree, so the data cannot be changed. The prediction graph for wrong or end of the lifecycle conditions using machine learning prediction (RF and LR) are shown in Figure 11.

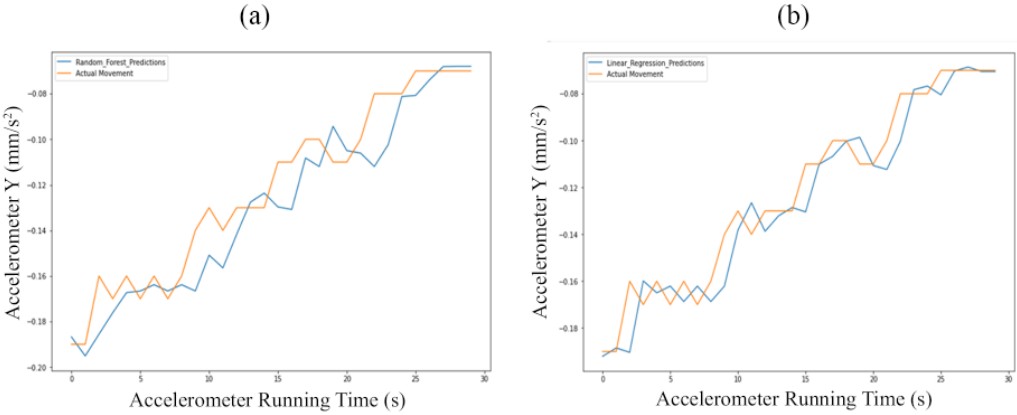

**Figure 11.** Data prediction is wrong/end of lifecycle condition using (**a**) RF regression method, (**b**) LR regression method.

The graph shows data prediction vs. real value of data from 26 data samples representing a cycle (same as a prediction of standard data). However, the graphic model is not a cycle because of function deterioration of the parts, such as wear or slack on the bushings/bearings, on the monitored components. The amount of data in the run-to-fail method is 28.12% from all data samples. The data range is between −0.20 mm and −0.08 mm. Figure 11a shows data accuracy using the RF regression method between 80–100% of each data. The accuracy of the RF regression prediction method in faulty or end-of-life cycle conditions is approximately 93%. The conclusion from both graphics or tables is the RF regression method that can be used in both linear and non-linear data predictions because of bagging from the decision tree diversification function using the random state function

in the algorithm. From Figure 11b, the LR prediction method for linear data is suited to predict the sensor movement in its function degradation. Data accuracy from each data is 84–100%, a 4% better individual prediction than the RF regression method. The LR prediction method's total accuracy is approximately 95% in faulty or end-of-lifecycle conditions. For implementation in the system, if the system accuracy for data prediction is high, the outcome must be higher than the algorithm with lower accuracy. So, the conclusion of the data analysis is written in Table 7.

**Table 7.** Vibration data prediction accuracy for each machine learning method in PdM implementation.

| Data Condition | Prediction Method | Accuracy | Total Accuracy within 20.000 Sample |
|---|---|---|---|
| Normal | Random Forest (RF) | 82% | 84% |
| Wrong/End Cycle | | 93% | 89% |
| Normal | Linear Regression (LR) | 23% | 59% |
| Wrong/End Cycle | | 95% | 94% |

Table 7 is calculated from all 20.000 sample average accuracy for each data using Equation (1), and it is concluded that the RF regression method has better accuracy than the LR regression method implemented in the PdM system because the training data model is not linear. The result of data prediction using the RF method agrees with Prihatno et al., where the RF method accuracy exceeded more than 80%, while the difference in the parameter is only in estimators (5000) and random state (42) [32]. The RF regression method has 29% more accuracy than the LR method in this case. The LR method has better accuracy than the RF method for failure detection by 2%. This statement is also made by Mattes et al., where the LR method showed the best performance shortly before failure on the test data (for semiconductors machines) [48]. Still, the accuracy of the overall system concludes that the RF regression method will less likely fail because it has only a 12% fail prediction. It is also shown in the MSE Results of each sensor—which can be seen in Table 8.

**Table 8.** Mean square error result comparison between RF regression and linear regression in normal and run-to-fail (RtF) methods in the sample thermocouple and accelerometer.

| Sensor Name | MSE (RF, Normal) | MSE (LR, Normal) | MSE (RF, RtF) | MSE (LR, RtF) |
|---|---|---|---|---|
| Accelerometer FPL1 | 0.02033 | 0.045498 | 0.02301 | 0.01140 |
| Accelerometer FPD1 | 0.02287 | 0.0486627 | 0.0229481 | 0.016253 |
| Thermocouple 1 | 0.018221 | 0.066091 | 0.012016 | 0.149674 |
| Thermocouple 2 | 0.019662 | 0.631192 | 0.111762 | 0.147899 |

The MSE of each data is close to 0, so the system's accuracy is better because of the MSE value.

### 3.3. Failure Model and Effect Analysis for the System

After we know the system's accuracy, the implementation of systems must be explained to the maintenance technician to do a specific action with the prediction result of the systems. One of the methods is FMEA, one of the most efficient low-risk tools for preventing and identifying problems for more efficacious solutions [49]. The FMEA table can be seen in Table 9.

From the FMEA analysis of the system, the technician can decide on the mechanical/electrical component fault, and the result can be seen in the implementation discussion.

**Table 9.** FMEA of the prediction of system failure.

| Comp. | Component Function | Functional Failure | Failure Mode | Failure Cause | Failure Effect |
|---|---|---|---|---|---|
| Accelerometer at filling pump drive 1 and 2. | Failure detection | The prediction graph cycle is smaller than the normal cycle. | 1.1. Prediction graphs cycle is smaller more than 0.5 mm | 1.1. There is loose bearing in the filling pump drive | Filling pump—no movement |
| | | | 1.2. Prediction graphs cycle is smaller from 0.2 mm to 0.4 mm | 1.2.a. There is a loose bushing in the filling pump drive | Unstable tube weight |
| | | | | 1.2.b. There is wearing in the filling pump drive parts (body or pen) | Unstable tube weight Filling pump overload |
| | | Prediction graph is not making a circle (increase/decrease) | 2.1. Prediction Graphs made an inclined graph | 2.1. The filling nozzle seal is already wearing | Leaked tube sealing |
| | | | 2.2. Prediction graphs made a declined graph | 2.2. Incorrect installation of bushings/body | Filling pump overload Leaked tube sealing |
| | | | 2.3. Prediction graphs are unstable | 2.3.a. There is a fault in the accelerometer | Prediction cannot be shown accurately |
| | | | | 2.3.b. There is improper wiring in the accelerometer | |
| Accelerometer at filling pump Lever 1 and 2. | failure detection | The prediction graph cycle is smaller than the normal cycle. | 1.1. Prediction graphs cycle is smaller more than 0.5 mm | 1.1. There is a loose bearing/bearing already wearing, in the filling pump lever | Filling pump no movement |
| | | | 1.2. Prediction graphs cycle is smaller from 0.2 mm to 0.4 mm | 1.2. There is a loose bushing in the filling pump lever | Filling pump overload Unstable tube weight |
| | | Prediction graph is not making a circle (increase/decrease) | 2.1. Prediction graphs made an inclined graph | 2.1.a. The piston nozzle seal is already wearing | Improper filling position |
| | | | | 2.1.b. Wearing plate (Parts) is already wearing | Filling pump overload Filling pump no movement |
| | | | 2.2. Prediction graphs made a declined graph | 2.2. The Piston Torpedo is already wearing | Improper filling position Unstable tube weight |
| | | | 2.3. Prediction graphs are unstable | 2.3.a. There is a fault in the accelerometer | Prediction cannot be shown accurately |
| | | | | 2.3.b. There is improper wiring in the accelerometer | |

<div align="center"><strong>Table 9.</strong> <em>Cont.</em></div>

| Comp. | Component Function | Functional Failure | Failure Mode | Failure Cause | Failure Effect |
|---|---|---|---|---|---|
| Accelerometer at coding cam Lever 1 and 2. | Failure detection | The prediction graph cycle is smaller than the normal cycle. | 1.1. Prediction graphs cycle is smaller more than 0.5 mm | 1.1. There is loose bearing in the coding cam lever | Leaked tube sealing |
| | | | 1.2. Prediction graphs cycle is smaller from 0.2 mm to 0.4 mm | 1.2.a. There is a loose bushing in the coding cam lever | Perforated tube sealing Wrinkled tube sealing |
| | | | | 1.2.b. The coding cam lever timing degree did not same as the main timing cam degree. | Wrinkled tube sealing Perforated tube sealing Leaked tube sealing |
| | | Prediction graph is not making a circle (increase/decrease) | 2.1. Prediction graphs made an inclined graph | 2.1. Coding cam Jaws position and settings are not proper | Perforated tube sealing Wrinkled tube sealing |
| | | | 2.2. Prediction graphs made a declined graph | 2.2. Coding cam Levers component wearing | Leaked tube sealing |
| | | | 2.3. Prediction graphs are unstable | 2.3.a. There is a fault in the accelerometer | Prediction cannot be shown accurately |
| | | | | 2.3.b. There is improper wiring in the accelerometer | |
| Thermocouple at hot air station | Temperature detection | The prediction graph cycle is smaller/larger than the normal cycle. | 1.1. Prediction graphs cycle is smaller more than 0.5 °C | 1.1. Thermocouple RUL is at its end | Perforated tube sealing |
| | | | 1.2. Prediction graphs cycle is larger by 1 °C | 1.2. Heater RUL is at its end | Leaked tube sealing |
| | | Prediction graph is not making a circle (increase/decrease) | 2.1. Prediction graphs made an inclined graph | 2.1. Thermo-control RUL is at its end | Perforated tube sealing |
| | | | 2.2. Prediction graphs made a declined graph | 2.2. Heater RUL is at its end | Leaked tube sealing |
| | | | 2.3. Prediction graphs are unstable | 2.3.a. There is a fault in the thermocouple | Prediction cannot be shown accurately |
| | | | | 2.3.b. There is improper wiring in the thermocouple | |
| | | The prediction graph is not detected | 3.1. Prediction is not showing | 3.1. Connection between thermocouple and systems is interrupted | Prediction cannot be done accurately |
| | | | 3.2. Prediction shows a straight graph at 0 | 3.2. There is a fault in the thermocouple | Hot air station is not working |

## 4. Implementation Discussion

PdM machine learning system using RF regression prediction has implemented in Norden NML 150 tube filling machine for three month trials, and its effectivity will be evaluated with:

1. Total machine UPDT comparison three months before and three months after;
2. Targeted machine UPDT comparison three months before and three months after;
3. Comparison of machine output three months before and three months after.

Figure 12 shows OEE data of the monitored tube filling machine three months before and three months after system implementation to observe system effectiveness. The OEE graphic represented each improvement aspect—lessening human error (such as setting errors in the filling and coding cams, better troubleshooting in the event of UPDT), predictive maintenance, and improving the quality control system.

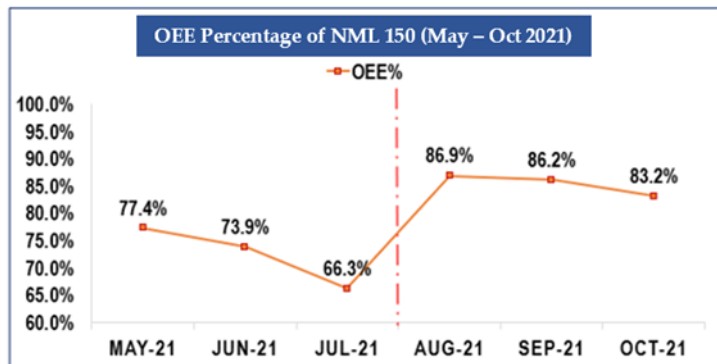

**Figure 12.** OEE of NML150 tube filling machine from May 2021–to October 2021.

From Figure 11, before system implementation, the OEE percentile of NML 150 could not reach 80% each month because of the low availability rate (AR) from many UPDT cases—from the human factor, machine breakdown, and other factors. After implementing the PdM system, the average machine OEE reached 85.35% because of predicted failure in the machine components, especially in the monitored parts stated in Section 2.1. Paolanti et al. also state that PdM is a good strategy for minimizing downtime and associated costs when dealing with maintenance issues [20]. Minimalized downtime shows in Table 10; there is a direct average OEE comparison between three months before and three months after PdM implementation, including its average performance rate (PR), quality rate (QR), and AR.

**Table 10.** System implementation effectiveness OEE measurements.

| Before (May–July 2021) | Measurement Parameter | After (August–October) |
|---|---|---|
| 75.10% | Average AR | 86.30% |
| 97.40% | Average PR | 99.60% |
| 99.00% | Average QR | 99.50% |
| 72.42% | Average OEE | 85.53% |

Overall improvement in the OEE case can be seen in Table 10, where the most significant increase was in the AR aspect, with an increase of 11.20% because there is a decrease in UPDT cases, frequency, and time. More details can be shown in the UPDT graph in Figure 13.

In Figure 13, the impact of system implementation is decreased in the UPDT case from machine breakdown and product quality issues because of unstable components/unpredicted breakdown in the monitored component. There are 14 cases of UPDT, with a reduction

percentage of 93.33% after implementing the system. From these reductions, some other factors are involved:

1.  Improvement in human resource/machine operator;
2.  Improvement in raw material qualities;
3.  Improvement in operational methods.

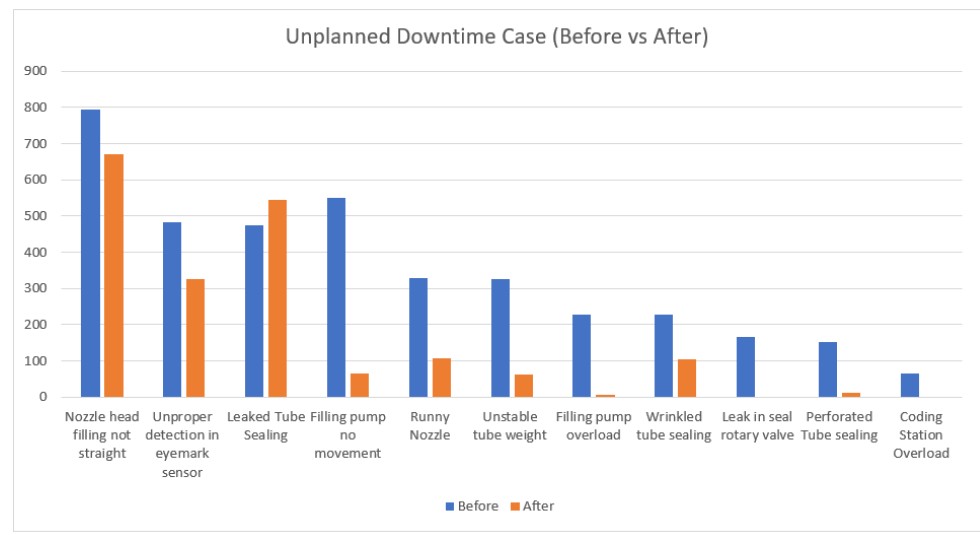

**Figure 13.** Update case reduction before vs. after system implementation.

These improvements are also a part of condition monitoring effects because the data shows a different trend for each problem, for example, each material/product type has its data characteristics. Each technician also has other characteristics when repairing the components for the incorrect data. There is a 1494-min breakdown reduction (the equivalent of 2 working days) with a 62.38% downtime percentage that can be decreased; details can be seen in Table 11.

**Table 11.** The effectiveness of downtime reduction in system implementation.

| Before (min) | Downtime Case | After (min) | Reduction (%) |
| --- | --- | --- | --- |
| 474 | Leaked tube sealing | 544 | +14.77% |
| 152 | Perforated tube sealing | 13 | −91.45% |
| 228 | Wrinkled tube sealing | 104 | −54.39% |
| 327 | Unstable tube weight | 62 | −81.04% |
| 425 | Filling pump no movement | 65 | −88.18% |
| 229 | Filling pump overload | 7 | −96.94% |
| 166 | Leak in seal rotary valve | 0 | −100.0% |
| 65 | Coding station overload | 0 | −100.0% |
| 329 | Runny Nozzle | 106 | −67.78% |
| 2520 | Total Monitored Downtime | 901 | −64.25% |

In conclusion, the PdM system using random forest regression prediction can be said to have succeeded in increasing OEE by 11.2% and reducing the downtime percentage of the monitored component UPDT by 64.25%

## 5. Conclusions

From the experiment, the actual ranges from the acceleration or vibration in the monitored component differ from 0.34 mm to 1.05 mm, and the temperature range varies from 0.02 °C to 1.93 °C. If the data occurs outside the said range, it may lead to component failure/component breakdown/quality issue of the product. The random forest regression prediction accuracy rate is better than the linear regression accuracy rate (88% to 59%),

which gained from the prediction data using the training data set. Implementation of the PdM system using the random forest regression prediction method effectively increased the OEE of the NML 150 tube filling machine with the said condition in Section 2.1. In addition, the cost-effective equipment works well with data acquisition and machine learning processing (predicting machine movements). The system successfully increased the machine OEE to an average of 11.31% three months after implementation (August–October 2021) and decreased the downtime to 62.38% from the monitored component. It also increased the machine output to an average of 2.20% for three months and repaired the product quality up to an average of 0.5% for three months. The increase in machine effectivity also means that the machine gets the right handling in carrying out maintenance activities and makes it easier for technicians to identify problems that occur in the machine. The conclusion is that the system can be implemented for tube filling machines and monitor the machine condition for proper PdM activity. Some future recommendations that can be done to improve the overall system are:

1. Improve microcontroller and hardware for data acquisition with a higher baud rate and sampling rate to have more accuracy of the data and a faster processing time in algorithm run;
2. Improvement in accelerometer and vibration sensor with a higher detection range and a higher sampling rate to make possible the fine-tuning of the system to obtain a faster failure prediction and a faster PdM action;
3. For further recommendation, the system can predict another component, product reject detection and prediction, minimize human error in the operational machine, and synchronize the environmental conditions with machine parts conditions.

**Author Contributions:** Conceptualization, methodology, software, formal analysis, investigation, resources, data curation, writing—original draft preparation, visualization, D.N.; conceptualization, methodology, writing, review and editing, supervision, H.S.; project administration, funding acquisition, D.N. and H.S. All authors have read and agreed to the published version of the manuscript.

**Funding:** This research was funded by the Institution of Research and Community Service, the Atma Jaya Catholic University of Indonesia in the 2021 grant.

**Institutional Review Board Statement:** Not applicable.

**Informed Consent Statement:** Not applicable.

**Data Availability Statement:** Not applicable.

**Acknowledgments:** We want to express our sincere gratitude to the Institution of Research and Community Service, the Atma Jaya Catholic University of Indonesia for funding this applied research.

**Conflicts of Interest:** The authors declare no conflict of interest.

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
