# Peer review of "Machine Learning Application Using Cost-Effective Components for Predictive Maintenance in Industry: A Tube Filling Machine Case Study"

_jmmp, doi:10.3390/jmmp6050108_

Round 1

Reviewer 1 Report

1-      Literature review is weak. The authors must emphasize and highlight the novelty of the research by providing a more detailed tabular literature review (from the latest and relevant research), elaborating the key parameters and approaches used by various researchers. The authors should review more research articles from 2020-2021-2022.

2-      Ishikawa Diagram – Cause and Effect Diagram must be included to elaborate the key parameters which effect the process. Must be supported by references.

3-      Design of Experiments: If any DOE approach is used, it must be mentioned.

4-      The choice for using random forest regression and linear regression method must be justified.

5-      Table 2. Downtime Analysis and affected parts classification, must be supported by references.

6-      Standards: If any standards were used in experimentation must be discussed too.

7-      Future Recommendations: conclusion must also provide the generic benefits and applications of the research carried out, furthermore, the limitations of the research, as well as the future research recommendations must also be included as well.

Reviewer 2 Report

The article discusses on the application of a random forest and linear regression models for the purposes of predictive maintenance of a tube filling machine.

However, both methods as well as the benefits stemming from the adoption of PdM approaches in manufacturing have been heavily investigated and are well established.

Moreover, scarce technical details are provided, for example key parameters of the two models, performance evaluation criteria, data volume considering the failures investigated in the article.

In addition, while the article does present some improvement in production wise KPIs, this is not associated to the performance of the models used nor justified.

Finally, the features/parameters used in each model as well as their correlation to each failure is not adequately discussed. The use of an FMECA or FMEA representing the failures investigated could support this. 

Hence, even though the article deals with a practical issue the context of a manufacturing system, the research-wise contribution of the article is unclear.

Reviewer 3 Report

The work presented is very interesting and the results provided are promising. However, there are some comments related to this work.

Introduction:

·         Full words that define abbreviations should be capitalized (line 32 - Prognostic and Health Management).

·         The Introduction section should be referred more at the challenges and needs of the production for new maintenance methods. You should specify your objectives and make clear to the reader the advantages of your approach.

·         Why did you choose to use Linear regression and Random Forest regression? Why these methods were the most suitable for your case study? Please explain more.

·        In the introduction, you make a literature review about PdM methods. So, the objective is to reference others’ works and methodologies to identify important gaps that the present approach deals with. It should be direct to the reader, at the end, what are the gaps and the key contributions of the present work in comparison to the identified gaps of other methods. So, you could make another section, a “Literature Review” section and add some papers in order to explain in detail the advantages of your approach. You can find below some relevant papers:

a.       https://doi.org/10.1371/journal.pone.0180944

b.       https://doi.org/10.3390/s21030972

c.       doi: 10.1109/SMC.2016.7844673

Materials and Methods:

·       Figure 6 is blur.

·       You should explain in detail the preprocessing steps of data and their effects on the results. Why the sampling amount is 20.000?

·     Maybe you should add an Implementation section in order to provide the details regarding the utilized hardware and software.

Results and Discussion:

·         Explain more about the classification you made. Why did you choose 2 categories (normal and run to fail)?

Reviewer 4 Report

Dear Authors,

Thank you for submitting your manuscript and sharing your work.  This is a very interesting topic and the approach, based on using low-cost sensing and computations is very attractive.  

Some aspects of your work were not clear in this version of the manuscript. The large number of observed downtimes need to be explained. To improve accessibility and clarity of your work, please consider providing more details on data acquisition and sampling. The machine learning work needs to be better specified: what was the model, how was the data split into training and test, etc.

Additional comments and feedback are provided as follows:

Figure 1 seems copied from a patent. It requires permission and needs to be explained in more detail.  The explanation in text, labeled (1)-(6), which annotate sections do not correspond to the figure. Please consider adapting and simplifying the figure and provide 1-to-1 correspondence with the text

Line 69-70: “The machine-learning algorithm must record the physical machine phenomena like vibration, temperature, etc”  

Strictly speaking, this sentence is not correct. The overall PdM system requires data collection of signals upon which ML operates (typically after the features have been extracted).

Line 73:

i                 The system will use two supervised machine learning methods to compare the effectivity of random forest regression and linear regression

Please provide additional specificity for the regression task, is it the prediction of time-to-failure, or something else.

Figure 2 shows astonishing large number of downtimes – approximately 5,000 downtimes in 6 months, or about 28 downtimes per day. How many machines were observed? Were these seeded failures? Please provide additional information to clarify the setup.  In addition, please label the y-axis to confirm that the numbers correspond to the observed downtimes.  Note that typically, very high rate of failures are handled by system redesign, not PdM (PdM systems are normally developed for assets with relatively low-frequency and high-severity of failures).

I struggled with the organization of Table 3. Please provide in the text more information.  What is Part Name, line-replaceable unit (LRU)?  What is amount, number of downtimes? If so, there are 18 (out of ~5 000 failures) – how were they selected?

Line 112: reference to Figure 4 comes out of nowhere, before it was introduced (and before Figure 3 is introduced).  Please reorganize the text.  Figure 4 has subfigure (a), (b), (c), (d) out of order. Please expand Figure 4 caption and explain what is in (a), (b), (c), and (d).

Figure 3 and the accompanied explanations (Line 128-137) are very good and informative. Please use this for other figures     

Figure 5: What was the empty bread board used, just for interconnections?

Please consider moving Table 5 up from Section 3 to Section 2. Please provide some rationale for selecting sampling rates for acceleration and temperature. Temperature is changes much more slowly than vibration and it is common to compute some features/condition indicators from vibration to synchronize the temperature and vibration information. Please provide more details on your approach. Did you just use raw data for both? Moreover, later on line 274, the manuscript reads “However, the thermocouple cycle runs faster than accelerometer data” This is very unusual and deserves more attention – please provide more details regarding your sampling choices.

Figure 7 – what are the units of vibration here? Normally, m/s^2 or g’s are used. Are these mm^2? Also, please consider using larger font for the labels.

Section 3.2

Please provide clearly what is your model, the inputs and outputs, what data was used for training and what data was used for validation (and test?)

For example, what input data was used to predict accelerometer output in Figure 9? Another accelerometer?

If possible, please consider sharing the data in the form of supplementary material or elsewhere to enable reproduceable science

Round 2

Reviewer 1 Report

The authors have revised the manuscript, however, it can still be further improved. 

Reviewer 3 Report

Dear Authors, thank you for considering my comments and improved the content of the manuscript. The paper is much better now. I have no comments.

Reviewer 4 Report

2nd review:

1.              For the details, we maximized to provide the data details in section 2 and sub-section 3. Please kindly check the sub-section.

The number of downtimes on a single machine is still unrealistic – and no clear explanation was provided: 5,378! – that is more than one downtime per hour if machine is operated non-stop (24x30x6 = 4,320)! This statement deserves more detailed explanation to be believable.

Sentence on line 125-126 is not finished

2.              Figure 1 has already been revised; please kindly check lines 50 -58. We have also added a new figure beside the patents - reference can be contained in References [20]. The patent picture has already been taken out.

- Thank you for these revisions – the new Figure 1 and the accompanied explanation are much improved

3.              The statement in lines 69 – 70 was corrected and erased from the article and stated in a different section explaining the machine learning algorithm.

Please be more specific and point to the specific lines in the text

4.         The additional specificity has been added in lines 118 – 123.

“The main goal of this research is to predict the time to failure of each observed component by feature extraction and machine learning, also measure each system's effectiveness and weakness to seek the most applicable method for vibration and temperature measurement

[33]     in tube filling machine applications because every machine part has its difference and characteristics, even for the exact variable measurement [34].”

Thank you for the clarification

5.         The explanation of downtimes is explained in lines 139 – 140. Figure 2 – now, Figure 3 has already been revised.

“The downtime occurred in the machine with low frequency (Less than three times/category) – and high repair time.”

The number of downtimes with more than one downtime per hour was not addressed (see the comment above)

6.         Table 3 was already revised. The explanation of table 3 can be checked in lines 152 – 155. “From Table 3, Component – measuring unit is the component names and its sensor to enable data acquisition, Amount is the amount of the components in one machine, and Downtime assessment is the classified downtime from Table 2, classified by the components. “

Thank you for the clarification

7.         Line 112 has been rearranged, the position of figure 4 – now figure 5 has already been fixed, the references of figure 3 – now figure 4 have already been added, and the setup of figure 5 has already been matched with figure 4.

Thank you for the revisions – they improved the manuscript. Please consider stating explicitly in the figure what (a)-(d) are instead of referring to Figure 4.

8.         Thank you so much for the positive comments; the same method has already been applied to all picture sections.

9.         The breadboard is used only for interconnection

It would be helpful if this was explicitly stated in the text

10.       Table 5 has been moved from section 3 to section 2. The cause of using acceleration and temperature – in the research, the approach is not to synchronize thermocouple and vibration but to use individual data of each sensor to predict each component, which can be seen in the FMEA table in Table 8.

Thank you for this explanation. However, some questions remain unclear.  Please explain (for each model) what is input and what is the output data (prediction).

Line 274 – now Line 343 – has a sort of different perspective and will be mentioned in the research in the sentences after line 343.

11.       The unit is mm2 because the processed data is in acceleration (sensor position). It was selected because of its simplicity and easier of feature extraction.  The Figures have already been revised.

This is simply incorrect: acceleration is not measured in mm^2. Mm^2 is the unit of area, not distance. Acceleration is measured in m/(s^2) and needs to be integrated with respect to time twice to obtain distance. Even in that case, the units would be in meters (or mm), but not mm^2! Please correct this error

12.       Validation data and prediction data come from the samples (20.000 samples) – and prediction is done by validating using the MSE method in section 3 sub-section 2, Table 8, by comparing data with similar research done by Prihatno et al. [32] – with a different application but using the same method. Data validation is mentioned in the line

In PHM and more broadly applied machine learning it is critical to clearly separate data for training a machine learning model and data used for validating the model (to avoid overfitting). Furthermore, it is normally practice to show the error associated with training machine learning models and compare them to that of validation data. Please explain how you separated your training from validation data. The provided explanation does not clarify sufficiently the question raised.

13. We apologize that we cannot provide supplementary materials because the programs have already been implemented, and they work as one of the company secrets.

“Data Availability Statement: Data Sharing is not applicable.”

-       Understood, data sharing is not always possible
